# Challenges of Pasture Feeding Systems—Opportunities and Constraints

**Barbara Wróbel** [1],*[ID]**, Waldemar Zielewicz** [2][ID]** and Mariola Staniak** [3][ID]

1   Institute of Technology and Life Sciences—National Research Institute, 3 Hrabska Avenue, 05-090 Raszyn, Poland
2   Department of Grassland and Natural Landscape Sciences, Poznań University of Life Sciences, Dojazd 11, 60-632 Poznan, Poland; waldemar.zielewicz@up.poznan.pl
3   Department of Forage Crop Production, Institute of Soil Science and Plant Cultivation-State Research Institute, Czartoryskich 8, 24-100 Puławy, Poland
*   Correspondence: b.wrobel@itp.edu.pl

**Abstract:** Grazing plays an important role in milk production in most regions of the world. Despite the importance of grazing, current trends in livestock farming in Europe are causing a decline in the popularity of pasture-based feeding of dairy cows. This paper aims to provide an overview of the challenges faced by the pasture feeding system under climate change. Grazing lands provide ecosystem services including regulation and storage of water flows, nutrient cycling, and C sequestration. Livestock grazing is the most important factor shaping and stabilizing pasture biodiversity. Some opportunities for pasture feeding are the health-promoting and nutritional qualities of milk and milk products, especially milk from pasture-fed cows. The beneficial effects of pasture feeding on animal health and welfare are not insignificant. Available organizational innovations can help better manage livestock grazing and, above all, better understand the impact of the grazing process on the environment and climate change.

**Keywords:** biodiversity; carbon sequestration; grazing systems; milk quality; welfare; virtual fencing





## 1. Introduction

Grazing has existed since the beginning of agriculture. According to the Food and Agriculture Organization, about 60% of the world's grasslands (slightly less than half of the world's land area) are covered by grazing systems. Grazing systems provide about 9% of the world's beef production and about 30% of the world's sheep and goat meat production. For about 100 million people in arid areas and probably a similar number elsewhere, grazing livestock is the only possible source of livelihood [1]. Pasture-based dairy production systems are mainly found in temperate regions, where grass is the cheapest feed used in milk production [2]. Pasture grazing can be used in feeding systems for dairy cows in other parts of Europe as well, but its importance is lower [3]. It is estimated that 98% of Irish and 92% of British dairy farms operate pasture-based systems, compared to only 20% in the Czech Republic, less than 10% in Greece, and virtually none in Bulgaria [4]. Even herds with access to pasture are typically kept indoors during the winter and around calving [5].

In general, for milk production, pasture grass is a higher quality forage than grass silage [6]. In temperate regions of Europe, grass growth is highly variable [7], varying among years [8], seasons, as well as regions [9]. It depends on many factors: pasture management, the sward renewal practices used, the level of fertilization, the course of weather conditions (e.g., precipitation, temperature, solar radiation) and soil type. The nutritional value of pasture sward varies depending on the season, growth stage, and age of regrowth.

The quality of pasture grass can be optimized through rational grazing and pasture management. For example, rotation length, biomass weight before grazing [10], and sward

height pre- and pos- grazing [11] can affect grass quality, as well as grass supply. A leafy sward in spring has a high nutritive value, while a sward at the reproductive stage in summer has a higher fiber content and lower digestibility [12]. The nutritional value of pasture sward also depends on their botanical composition. Swards with a significant proportion of legumes often have a higher feed value than grass-only swards [13,14].

Rationally used pasture provides grazing animals with high-quality roughage, containing mainly energy, protein, macro- and microelements, and vitamins [15,16]. In addition to valuable species of grasses and legumes, the composition of pasture sward includes dicotyledonous plants called herbs. They contain many valuable biologically active substances such as tannins, flavonoids, saponins, pectins, terpenes, alkaloids, phenols, as well as essential oils [17,18]. These compounds have positive effects on cattle gastrointestinal function and health (antioxidant and antiparasitic effects, enhancement of the immune system) [19,20], and the quality of beef and dairy products [21].

Livestock grazing influences pasture biodiversity, particularly the botanical composition of plant communities [22–24], as well as the quantity and quality of forage produced [25], the dynamics of sward regrowth [26], the variability of species occurrence and contribution, and the landscape that pastures create [27,28]. Pasture use contributes to a rich diversity and variability in the vegetation cover maintaining the maintenance of all forms of biodiversity [29–31]. However, the diversity of plant communities created by grazing by animals can change depending on environmental conditions, including regional climate variability [32], grazing intensity [33], and soil nutrient availability [34]. Grazing lands also provide ecosystem services including regulation and storage of water flows [35], nutrient cycling, and C sequestration [36,37].

However, in addition to the advantages of pasture feeding, related to the ability of cattle to consume good quality roughage, the positive impact on animal welfare, and the quality of animal products, there are unfortunately also some difficulties. It was calculated that a grass-only diet can support milk production levels of 22 to 28 kg cow$^{-1}$ day$^{-1}$ [38]. Therefore pasture-based feeding of dairy cows is used to a greater extent only on farms with average productivity. Although green fodder completely covers energy and protein requirements, there may be periodic minor deficiencies and fluctuations in individual components due to grazing dates and weather conditions [39].

Despite the importance of grazing, current trends in livestock farming in Europe are causing a decline in the popularity of pasture-based feeding of dairy cows [40]. The reasons for this vary, mainly due to the different farming systems that typically exist in each region [41]. In Central Europe, dairy farmers are under pressure to maximize milk production [42]. For this reason, dairy cows are often not grazed but are kept indoors for their entire lives [43] and fed mainly silage and concentrate feeds [44]. Pasture feeding, in some respects, is more time- and labor-consuming than keeping cattle indoors or in loose housing systems. Grazing livestock on pasture also involves costs associated with the purchase and installation of pasture fencing and the designation of paddocks [45,46]. In addition, electric fences and metal mesh structures can disrupt the natural landscape and prevent the migration of some wildlife species [47,48].

Agriculture is significantly influenced by climate change, while also being a driver of climate change through the release of greenhouse gases (GHGs) into the atmosphere. Therefore, reducing the carbon footprint generated by agriculture is key to reducing climate change. At the same time, agriculture and forestry can remove $CO_2$ from the atmosphere. Particularly pastures play a significant role in climate change due to massive stores and fluxes of C [49].

This paper aims to provide an overview of the challenges faced by the pasture feeding system under climate change.

## 2. Challenges from the Environment

The EU's common agricultural policy and environmental legislation, such as the EU Nitrates Directive, are putting pressure on farmers to ensure that EU milk production is

both economically and environmentally sustainable. Global food production accounts for 30% of anthropogenic greenhouse gas emissions, causing climate change [50]. Ruminant animals are a major contributor to these emissions, particularly through belched methane, which is produced during the intestinal fermentation process [51,52]. The detrimental effects of methane in the atmosphere have led to movements and social initiatives to reduce animal production to ensure a rapid reduction in atmospheric methane emissions as the fastest route in mitigating progressive climate change [53].

### 2.1. Carbon Sequestration

Climate change has created the need to find solutions that can counter greenhouse gas emissions. One of these is the development and maintenance of natural carbon (C) sink habitats, such as forests and permanent grasslands.

Grasslands are one of the most important biomes on earth [54]. They are estimated to contain about 30% of the world's carbon stocks [55]. Their role in carbon sequestration is important, as it is estimated that it could be around 590 billion tonnes of carbon dioxide emissions [56]. According to Bai and Cotrufo [49], the achievable soil organic carbon sequestration potential in global grasslands is 2.3 to 7.3 billion tons of carbon dioxide equivalents per year ($CO_2$ e $year^{-1}$) for biodiversity restoration, 148 to 699 megatons of $CO_2$ e $year^{-1}$ for improved grazing management, and 147 megatons of $CO_2$ e $year^{-1}$ for sown legumes in pasturelands. However, the intensity of sequestration and carbon storage in grazing lands depends on the climate, location in the landscape, land use, as well as plant community type [57,58]. Studies on the effects of grazing intensity on carbon sequestration showed that intensive grazing mostly reduced C storage, while low- to moderate-intensive grazing balanced the amount of C sequestration with livestock production [59]. Adaptive Multi-Paddock (MPA) grazing [60] as well as "rotatinuous stocking" [61] has been shown to sequester more soil carbon than the traditional continuous and rotational grazing used in the past by many farmers. Research by Funakawa [62] indicates that high soil acidity can inhibit soil microbial activity and increase soil organic carbon accumulation. This research suggests that organic carbon storage and the soil biodiversity-carbon relationship are controlled by many interrelated processes and complex plant–soil feedbacks.

Over the past decade, the area of grassland has been shrinking while the area of arable land has been increasing, suggesting a continued conversion of grassland to arable land [63]. It is estimated that about 20% of the world's grasslands have been converted to arable land, leading to a loss of up to 60% of soil carbon stocks [64,65].

Another problem is the progressive degradation of grassland defined as a long-term decline in ecosystem function and measured in terms of net primary productivity [66], which results in the loss of carbon stocks in the grassland ecosystem. Stopping the processes of grassland degradation and its conversion to arable land would conserve grassland soil carbon stocks. Reversing the practices that have led to grassland degradation could increase ecosystem carbon stocks by sequestering atmospheric $CO_2$ in grassland soils. Estimates of the carbon sequestration potential of pastures are not well understood [67]. The lack of certain data and knowledge on this topic is due to several limitations. First, pasture data are collected on a smaller scale than data on forests and croplands [68]. Second, information on how pasture management affects soil carbon stocks is limited to only certain regions of the world.

### 2.2. Air Protection

Livestock contributes to greenhouse gas emissions in the form of methane ($CH_4$) from enteric fermentation, nitrous oxide ($N_2O$) from the use of nitrogen fertilizers, and $N_2O$ and $CH_4$ from animal excreta management and deposition. Carbon dioxide ($CO_2$) is also produced from the use of energy and fossil fuel on farms [69]. However, the amounts are not as significant as previously assumed. For example, in Germany (the largest producer of milk and pork and the second largest producer of beef in the EU), agriculture was directly responsible for 8.2% of all GHG emissions in 2020 [70]. Moreover, agriculture is the only

sector that can remove GHG emissions from the atmosphere. $CO_2$ in the atmosphere can be placed back into soils through natural grazing [71].

The amounts of waste production and gas emissions ($CH_4$ and $CO_2$) are largely determined by the intensity of animal production [72]. Studies show that the more intensively animals are reared, the lower the unit environmental load. The reason for this is more intensive feeding (amount of protein and energy supplied per unit of feed), better utilization of the animal's genetic potential for growth (higher efficiency of digestive processes and muscle tissue growth), and thus shorter rearing times. In dairy cattle, on the other hand, it has been calculated that a three-fold increase in milk production (from 4000 to 12,000 kg per year) decreases the methane emission per kg of product by only 48% [73]. The decrease in methane production following the increase in milk production is due to a change in nutrition, involving a reduction in the proportion of roughage in favor of concentrate in the diet, and a consequent change in the rumen fermentation profile (acetate fermentation decreases and propionate fermentation increases). To improve the climate, it would make sense to selectively reduce livestock production, above all those with high methane and $CO_2$ emissions.

For reducing greenhouse gas emissions, grazing, and feeding, management strategies are essential. Rational grazing management harvests higher-value forage, allowing more efficient use of nutrients, which increases animal productivity [74]. Additional opportunities to reduce gaseous emissions associated with ruminant food production include improved livestock health, fertility, and productivity. Increased productivity per head in the form of milk production or faster meat gains per day can reduce the number of animals required to maintain a given level of food produced. Another way is to use feed additives to reduce the intensity of methane production in the gut and to improve the quality and digestibility of grassland feed, which will ultimately reduce methane production [75].

*2.3. Environment Protection*

Grazing has several effects on the environment but the most evident is reduction of nutrient loss. Reduced grazing results in reduced mineral losses and less imbalance between mineral inputs and mineral emissions. This is especially true for nitrogen and phosphorus compounds. When animals graze on pasture, manure ends up in a small area of pasture where nutrients cannot be fully utilized, and therefore losses are more likely. It is estimated that keeping cows indoors year-round improves nutrient use efficiency and reduces the need to purchase mineral fertilizers by about 50 kg ha$^{-1}$ per year compared to the grazing system [76]. In addition, grazing affects the type of nitrogen loss.

Nitrogen cycling in a grazing system is influenced by the diet of the grazing animal and the distribution of ingested N within the animal [77]. In general, 75 to 95% of ingested N is returned to the soil, and about 70% of this N is excreted in the urine [78]. Thus, the main source of nitrate leaching from pastures is livestock urine, and during grazing relatively large amounts of nitrate may be leached [79]. Although leaching and volatilization (of animal urine) represent the main pathways of N loss in extensive grasslands, denitrification is a major pathway of loss in fertilized pastures which have a larger pool of readily metabolizable N sources [80].

By contrast, keeping cows indoors all year and manure collecting and spreading it on the land result in more ammonia volatilization. This ammonia volatilization may be partly reduced by adapting the feed strategy (less protein in the ration) [81]. Keeping cows indoors all year may cause higher energy use and hence the $CO_2$ emissions because of the need for machinery use. The grazing system does not affect methane emissions from grasslands themselves [82]. Keeping cows indoors all year, however, may lead to more methane emissions than grazing [83].

*2.4. Protection of Biodiversity*

Permanent grasslands, including pastures, are characterized by a diverse species composition, which is the result of interspecific competition modified by human activity.

The most important role is played by habitat conditions and the method and intensity of agricultural use, which determine the type of sward and its natural values.

One of the characteristics of high nature-value grasslands is their species richness. However, maintaining this richness is not a priority on farms focused on intensive livestock production. On such farms, the most important thing is to obtain large quantities of valuable fodder, so the composition of the sward is generally characterized by a poor species composition, dominated by noble grasses, with a proportion of legumes and a small number of herbaceous species, more of which are regarded, from a production point of view, as weeds, i.e., a symptom of degradation. On the other hand, the maintenance of a high species diversity is favored by extensive grazing and the limited use of pratotechnical treatments.

One of the most important factors shaping and stabilizing valuable communities in permanent grasslands is sustainable grazing, which is sometimes used as a protective treatment to stop the encroachment of secondary succession [84–86]. Grazing has a beneficial effect on floristic composition and sward structure, contributing to the maintenance of an open pastoral landscape and preserving habitat species richness. Vegetation structure in pastures is the result of environmental processes and factors determined by grazing animals, i.e., their food requirements and preferences and grazing intensity. In addition, moderate trampling of the sward by grazing animals leads to localized damage to the sward and the formation of germination gaps in which seeds can germinate and, in addition, allow light access to the lower parts of the sward and better development of light-loving species [87]. In addition, animals carry diaspores of plants, facilitating their reproduction and spread. Animal droppings provide nutrients point-wise to the habitat, leading to the production of numerous microhabitats [88].

Due to the habitat conditions, grassland communities situated on fairly rich mineral soils with low groundwater levels are mainly suitable for grazing. In the UK, grazing is a valued form of grassland conservation on mineral, wet, fluctuating wet, and boggy soils [89]. Habitats developed on moist organic soils should not be grazed too extensively because of the risk of more rapid sward and topsoil destruction compared with bog habitats located on the mineral ground.

Studies show that in some habitat types, extensive grazing after harvesting ha a beneficial effect on the floristic diversity of the meadow [86,87,90–92]. A periodic change of use, in particular the introduction of grazing every few years in some types of hay meadows, can also be beneficial. This has a beneficial effect on the sooding of the sward. In addition, grazing animals, through gnawing and trampling, reduce the expansion of undesirable species that did not disappear under the influence of mowing alone, e.g., *Molinia caerulea* (L.) Moench [93–95].

The intensity of plant grazing is directly influenced by the weight of the animals grazing a particular area at a given time (the so-called pasture load). Over-intensive grazing (too high a stocking rate or load), which results in almost the entire sward being grazed, can lead to the elimination of rare plant species and impoverishment of the habitat structure [96]. It can also lead to soil exposure, resulting in soil drying [87]. In turn, overextensification of grazing contributes to disturbances in species composition and encroachment of expansive vegetation. The potential of the pasture is also not fully exploited. Too low a stocking rate and pasture load lead to selective grazing of plants and an increase in the proportion of herbaceous vegetation, with a consequent deterioration of the forage and wildlife value of the habitat [87]. In turn, complete abandonment of grazing and lack of use contribute to changes in habitat conditions that initiate secondary succession. The habitat loses valuable and characteristic plant species, and the expansion of competing species begins with ruderal, herbaceous, alien, and forest vegetation [97]. This prevents the development of other plants and provides a refuge for pests of neighboring crops. This phenomenon is particularly exacerbated in areas, where agricultural production encounters environmental barriers, such as mountainous areas or unregulated river valleys. Fallowing or overgrazing are the main threats to extensive pastures, especially species-rich thermophilous grasslands, whose unique flora and fauna depend on extensive grazing. Inappropriate use results in

their conversion to poor grasslands and, in addition, the breeding grounds of birds such as lapwing, ruff, and stonechat are destroyed. A decline in the number of these species, caused by the disappearance of traditional agriculture, is observed both in Poland and in the rest of Europe [98]. Conservation practice has shown that the most effective tool for active protection of these birds is extensive grazing by cattle and horses, which creates a favorable mosaic of lower and higher vegetation and covered soil [98].

Each species of grazing animal behaves and affects the vegetation in a distinctive way [87,99]. Cows, horses, and sheep are the most commonly grazed animals, less frequently goats, geese, and other animals. The most versatile and common is cattle, known for their high tolerance to forage consumed and too moist habitats and sodden ground. In addition, the dietary preferences of cattle result in some plant species gnawing low and others only in the top parts, which affects the regrowth and botanical composition of the sward. Horses are more demanding than cattle, but as the growing season progresses they begin to reach for species not previously eaten. They trample the sward and soil more strongly than cows, which can be used in habitats where coarse vegetation needs to be disposed of [100]. Sheep, on the other hand, are animals that selectively take up plants and bite them low [101]. In the mountains, pastures are quite commonly grazed, contributing to the conservation of habitats and rare and protected species [102]. On pastures located in areas more difficult to access by larger ruminants, goats are often used in addition to sheep [103]. They are considered to be not very selective but prefer legume plants and bark, as well as leaves and shoots of trees and shrubs. Mixed grazing of several animal species simultaneously can also be used.

Extensive grasslands used to be very common. They can be located in wet habitats, moderately water-rich areas, and even on poor and dry soils. In some areas of the world (e.g., Australia), grazing cattle, sheep, or goats is the only way to utilize vast areas of pasture in arid regions that are often too barren and vegetation-poor for intensive animal breeding [104]. In Africa, on the other hand, extensive semi-arid savannah pastures are used by various wild ungulates in addition to cattle. Excessive stocking of such pastures with cattle is detrimental to wild herbivores. Lower cattle stocking rates and extensive grazing have been more beneficial to vegetation and wild herbivores and to maintaining satisfactory livestock productivity [105].

Grazing on pastures located in moist habitats improves animal production rates, as well as sward persistence and flora biodiversity [106]. Lowland pastures include grasslands developed in wet habitats with mineral soils (riparian and oak-hornbeam). They are characterized by an increased proportion of *Juncus effusus* (L.), *Juncus articulatus* (L.), and also *Ranunculus repens* (L.) and *Agrostis stolonifera* (L.) [107]. The most valuable lowland pastures are located in river valleys, on mineral soils characterized by high resistance to gnawing and trampling. The vegetation forms a relatively low and dense carpet-like sward. They are found in intensively grazed areas of Europe in relatively fertile locations and on moderately moist soils. Lowland pastures are threatened primarily by intensification of use and the increased burden of long-term grazing. Overgrazing leads to the conversion of extensive pastures into poor and low-yielding grasslands [108,109].

Traditions of sheep and cattle grazing are strongly linked to mountain and foothill areas. In the past, grazing in mountain areas was the primary form of utilization of the grasslands above the forest boundary. Nowadays, grazing on mountain pastures, which when not used, are overgrown with bushes and trees, has a special natural role. Many extensive mountain pastures are characterized by a high richness of flora, with a high proportion of *Cynosurus cristatus* (L.) and *Festuca rubra* (L.) [110]. A slightly different type of *montane* community is poor extensive pastures abundantly overgrown by *Nardus stricta* (L.), *Arnica montana* (L.), and *Carlina acaulis* (L.) and various orchid species.

Grasslands with a predominance of *Armeria maritima* (Willd.) are typical pastures located on dry and sandy soils in river valleys and on strongly drying riparian sites in summer. Characteristic species growing in such pastures are also: *Thymus serpyllum* (L.) and *Dianthus deltoides* (L.). *Pilosella officinarum* (Vaill.) and *Galium verum* (L.) can also often

be found on such sites. Among the grasses, narrow-leaved species predominate, such as *Festuca ovina* (L.), *Festuca rubra (L.),* and *Agrostis capillaris* (L.) [111,112].

Pastures and thermophilous grasslands are among the rarest plant communities. Their habitat conditions and species composition are similar to steppe communities. They occur, above all, in the upland areas of Central and Eastern Europe and the valleys of large rivers on their steep southern slopes. A characteristic feature of these habitats is the high abundance of calcium carbonate ($CaCO_3$) in loess and rendzina soils. The traditional use is extensive grazing by cattle, sheep, goats, and sometimes horses. The main natural value of thermophilic grasslands and pastures is the high plant species richness. Among the grasses, *Stipa joannis* (Čelak.) and *Stipa capillata* (L.) are the most abundant in the pasture sward. Species of small sedges such as *Carex humilis* (L.) and *Carex supina* (Willd.) are also encountered. The most abundant herbs are *Scabiosa canescens* (Waldst. & Kit.), *Prunella grandiflora* (L.) Scholler. and *Salvia pratensis* (L.) [113,114].

A major threat to meadows and pastures is posed by alien invasive species, which are becoming increasingly widespread and are encroaching primarily on unused or overly extensively used habitats. The encroachment of alien species is a threat to native flora. Invasion by alien species can result in a change in the species composition of the community or the complete elimination of native species, leading to the formation of communities with a dominance of alien species. The displacement of native species poses a threat to them and may lead to a reduction in the gene pool of the natural vegetation [115].

## 3. Impact of Grazing on Animal Productivity and the Environment

The production results of grazing animals depend on the stocking method and management of grazing. In general, there are two stocking methods, i.e., rotational and continuous stocking, among which different modifications are used depending on the various factors. Farmers have different considerations when choosing a stocking methods. In their choice, they may take into account the impact of grazing on yield and forage utilization, but also many other factors such as environmental impact, animal welfare, and other aspects. In some countries, legislation is a decisive factor. Grazing methods and pasture organization should be optimally adapted to the possibilities and specific characteristics of the farm.

### 3.1. Stocking Methods

The cheapest stocking method for cattle, mainly used on extensive pastures, involving continuous stocking of the sward, from spring to autumn, over the entire pasture area, is free grazing. The supply of forage depends on the season, with an over-supply of forage in spring, while there can be a periodic shortage of forage in summer when rainfall is scarce. Carrying out maintenance and rational fertilization in this stocking method is very difficult. Pastures under this stocking method are prone to a more rapid degradation process, involving the disappearance of valuable grass and legumes and the development of weeds [116].

All modern intensive stocking methods use the principles of rotational stocking [117]. Rotational stocking involves the frequent movement of livestock through a series of pasture subdivisions called paddocks [118]. Rotational stocking has many potential economic and environmental advantages [119,120] such as increases herbage production for livestock [121,122] and improves animal production [123], prevents overgrazing, and reduces soil erosion [124]. Rotational stocking has been found to improve soil microbial activity [125], which may promote greater stabilization of organic matter [126]. Moreover, rotational stocking results in fewer herd health problems and many others. Jordon et al. [127] provided empirical confirmation of the mechanisms by which rotational stocking and increasing sward biodiversity through the inclusion of perennial herbaceous plants (herbaceous strips) can increase forage production and animal growth rates. However, some studies have shown that rotational stocking does not provide any unique ecological or agricultural benefits compared to continuous stocking [120]. But, more important than the stocking method is the grazing intensity which is thought to have a major impact

on soil organic carbon storage and soil quality indicators in grassland agroecosystems. Moreover, soil improvement resulting from intensive rotational stocking does not occur rapidly [122,128]. It takes three to five years to start seeing beneficial changes in vegetation cover and soil microbial activity [129]. In general, intensive rotational stocking is more likely to be successful in areas with higher rainfall [106]. In northern Spain, where the average annual rainfall is 1000 mm, intensive rotational stocking by sheep resulted in higher forage production and increased carbon sequestration [130]. In more arid areas, intensive rotational stocking with frequent movement of animals can result in reduced weight gain [104]. Rotational stocking also has some disadvantages. It requires more fencing and labor (an effective alternative to traditional fencing is virtual fencing). It may result in soil compaction and degraded water quality if livestock is not moved regularly, as well as may increase internal parasites in irrigated rotational pastures.

To some extent similar to rotational stocking is guarded grazing, where the herd is supervised by a shepherd. This involves the animals returning after a certain period to areas previously grazed. This stocking method is practiced mainly in the mountains when grazing sheep and can be of great importance in naturally valuable areas as a factor stimulating the increase of their biodiversity and the preservation of naturally valuable areas [131].

A very efficient method of rotational stocking is tethered grazing (staking the animals). The great advantage of this method is the possibility of feeding each animal individually, easily regulating the amount of forage available to it. This stocking method is mainly used on farms with few cattle or horses. Staking causes some inconvenience due to the need to move the stakes and water the animals (bringing them to the watering hole or bringing water to the pasture). When grazing in this method, the animals have limited movement and, if left in the same place for a long time, they may eat the sward too intensively, which can cause damage by too much trampling or low biting.

Continuous stocking is the least controlled of the stocking methods. It consists of grazing the sward, over the whole or partially regulated area of the pasture, from spring to autumn in a slow method. The basis of this method is the control of the height of the grazed sward. As the sward is grazed and the rate of increase decreases, additional spare areas are incorporated into the grazing area to provide a reserve of feed for the whole herd. This method is often used by farmers with relatively large pasture areas and low numbers of livestock. Continuous stocking usually results in lower productivity per animal and lower output per unit of land. This stocking method is applied for animals that do not require high maintenance, such as sheep, dry cows, growing heifers, and low-milking cows. It requires lower amounts of labor, fencing, and water sources. The animals selectively graze the most palatable forage, which generally increases gains per animal. Selective grazing reduces total pasture productivity and leads to overgrazing in some parts of the pasture. Forage use can be improved by varying the stocking rate or temporarily fencing off part of the pasture for herbage harvest ("buffer" system).

Ultra-high Stock Density (UHSD) or commonly known as "Mob Grazing" or "Flash Grazing" is a short-duration, high-density grazing with a longer than usual grass recovery period. It has been proposed as a way to increase soil carbon storage and range quality. This system has been adopted in the USA, Canada, and the UK [132] for intensive grazing of cattle, sheep, and goats. High stocking densities result in a high fertilizing effect of the manure left on the small pasture area. The high stocking rate of the pasture means that much of the plant biomass is trampled by the hooves of the animals into the ground to form mulch that protects the soil from erosion. Plant residues and animal excreta contribute to an increase in organic matter content and improve soil nutrient abundance, which positively influences soil microorganisms and stimulates plant production [133]. However, some studies show a negative effect of mob-grazing on soil organic matter and other desirable properties of pastures [134]. According to [135], high densities of livestock increase soil C in warm-season grasses and decrease soil C in cool-season grasses. Moreover, mob-grazing can cause soil erosion by enhancing soil compaction, which has a negative impact on soil

water infiltration and plant growth [57,136]. There is a need for a better understanding of the mechanisms by which mob-grazing may positively or negatively influence soil C storage and vegetation [120].

Adaptive Multi-Paddock (AMP) grazing has been developed as a conservation-oriented grazing management approach for improving the ecological function of grazed ecosystems by continuously adjusting the number of grazing animals and the duration of herbivory in response to changes in forage availability [137,138]. Multi-paddock grazing management has been recommended since the mid-20th century as an important tool to adaptively manage rangeland ecosystems to sustain productivity and improve animal management. AMP grazing employs multiple paddocks per herd to enable short grazing periods leaving sufficient post-herbivory plant residue for regrowth, and long recovery periods to accommodate seasonal variation in plant regrowth [139]. It was found that, based on restored soil health, water conservation, and improved ecosystem services, AMP grazing was superior to heavy continuous grazing [121]. Similarly, it was found by Hillenbrand et al. [140] that AMP grazing improves the forage biomass, water infiltration, and total soil carbon concerning heavy continuous grazing [60]. It was confirmed that long-term AMP grazing improves streamflow, water balances, and water quality at the ranch and watershed scales [141–143]. AMP grazing also increases net primary productivity, soil C and N, and reduced C losses in runoff and sediment [144].

Alternative pasture management which can be used to increase ruminant performance and reduce gastro-intestinal nematodes is mixed grazing. In Germany, sheep and goat grazing is used to rehabilitate areas over-exploited by intensive cattle grazing [145]. This is due to the different dietary preferences of the different animal species. Sheep and goats eat woody and low-value plants that are avoided by cattle. In addition, sheep are less picky about the plants growing next to the dung left by cattle, which contributes to the increased forage used. Mixed grazing, compared to grazing only one animal species, not only allows better utilization of the sward [146] but increases the biodiversity of the sward and soil bacterial flora [147,148], arthropods, and birds [149]. Research in Ireland on cattle and sheep herds showed that sheep follow grazing after cattle promoted a higher proportion of clover in the sward and a greater number of clover volunteers from seed not digested by cattle and sheep [146]. An additional benefit of such grazing is less sward damage, reduced invasion by animal parasites and the emergence of more beneficial plant-pollinating insects [150].

The silvopastoral grazing system (SPS) involves grazing animals in wooded areas, traditional orchards, and groves (Figure 1). It is a system of short rotation of animals staying in tree-lined pastures. This system is commonly found throughout the world. Trees and bushy vegetation provide shelter for the animals, but can also provide food [151]. Husak and Grado [152] found that this grazing system contributes to sustainable livestock production and increases the productivity, profitability, and viability of area use. There is considerable evidence that SPS can increase production efficiency, increase carbon sequestration, and improve N cycling on land used for livestock production [153]. Other advantages of this system are the restoration of uncultivated land to agricultural production, low labor inputs compared to intensive production, improved welfare of beef cattle (minimal stress on the animals), and high-quality beef sold as an organic product. This is supported by a study by Skonieski et al. [154], according to which the SPS improved the welfare of grazing Jersey cows, as evidenced by an improved physiological response to heat stress, increased grazing time, and reduced standing time (resting + ruminating) compared to cows grazing on conventional pasture.

Sometimes, innovative producers are grazing sheep in the areas occupied by farms with solar PV panels. These surfaces also need tending, mowing, and biomass removal, so sheep and goats are increasingly being used for this purpose. Grazing sheep under such panels is possible without special modifications to the photovoltaic installations. Grazing cattle under solar panels requires stronger support poles and panels installed higher off the ground [155,156].

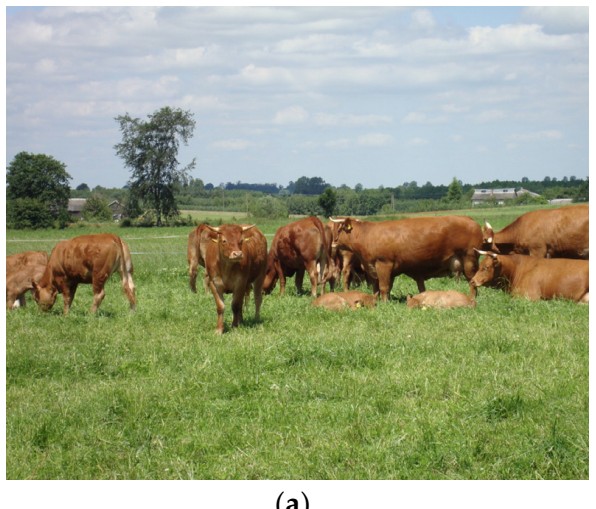
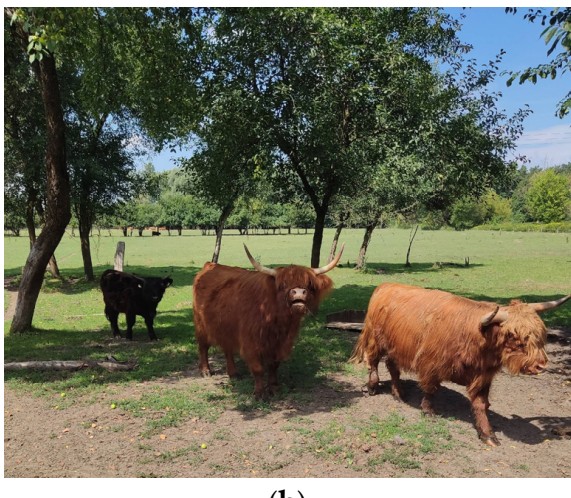

(**a**)　　　　　　　　　　　　　　　　　　　　　　　　　(**b**)

**Figure 1.** Feeding systems (**a**) continuous grazing; (**b**) Silvopastoral grazing system in wooded areas (Fot. M. Staniak).

### *3.2. Organisational Innovations in Grazing*

In addition to the advantages of pasture feeding, related to the possibility for cattle to consume good quality roughage, the positive impact on animal welfare, and the higher quality of animal products, there are unfortunately also some disadvantages (Table 1). Pasture feeding is undoubtedly more time-consuming and labor-intensive in some respects (animal monitoring and grazing management) compared to keeping cattle in alcove systems. Grazing animals on pastures also entails costs related to the purchase and installation of structural elements for pasture fencing [45]. But undoubtedly grazing is the cheaper way to feed domestic herbivores, and structures to maintain animals indoors have a higher cost than to maintain in pastures even with different paddocks.

**Table 1.** Advantages and disadvantages of pasture feeding.

| | Advantages | Disadvantages |
|---|---|---|
| Environment | Lower energy consumption Less $CO_2$, ammonia, and methane emission | More N losses (nitrate leaching denitrification, NO emissions) |
| Soil | Soil quality improvement (Mob grazing and AMP) | Increase in soil compaction |
| Biodiversity | Rational grazing increases biodiversity | Overgrazing reduces biodiversity |
| Sward quality and availability | Grazing reduces the need for sward renewal | Variability in forage availability and quality during the grazing season. Relatively large fluctuations in the composition of the ration |
| Animal health and welfare | Lower risk of various diseases Possibility of natural behavior | Higher risk of infection with internal parasites The risk of stress caused by weather factors |
| Milk quality | Better milk and dairy products quality | More variable milk quality due to the variability of grass supply and quality |

Innovations and modern technologies applied to grasslands aim to facilitate and improve the process of rational grazing management of livestock herds. Increasingly, modern computer technology is being used in grassland management (Table 2). Drones, specialized sensors, and sensors are being used to make more efficient use of pastures and to monitor the availability of forage for cattle on an ongoing basis (Figure 2). On an ongoing, real-time basis, daily grass growth, the amount of forage consumed by the animals, and the condition of the soil can be monitored and determined. It is also possible to locate animals

in the pasture in real-time and assess their activity, behavior, and the activities they are performing at the moment [157].

### 3.2.1. Virtual Fencing

In countries where the pasture feeding system is well developed, virtual pasture fences are used, delimiting the area to be grazed. When the animal approaches the virtual zone, it is given an audible signal, which tells it to stop. The farmer remotely—in the office, at the computer—determines the area to be grazed, over which the cows move independently [157]. Virtual fencing devices use an algorithm that combines GPS animal positioning with animal behavior to implement the virtual fence [158,159]. Like conventional fencing, virtual fencing is used to provide a boundary to the grazing area to deter animals from moving further, but unlike conventional fencing, it does not create a physical barrier [160]. With a virtual fence system, animals learn a virtual barrier not to cross by associating a sound stimulus with an electrical stimulus. When approaching the fence boundary, a warning acoustic signal is triggered and the electrical impulse stimulus from the collar is only produced as a punishment if the animal continues to move forward. If the animal turns away or stops at the audible signal, the electrical impulse stimulus is not initiated by the collar. Cattle have been shown to learn this association easily in several trials; however, there is a high variability in learning and behavioral responses between individuals [161–163]. Virtual fencing is highly useful and has great potential for controlling sheep distribution during grazing, but the development of virtual fencing technology for sheep grazing is still less advanced than for cattle [164–167].

### 3.2.2. Automation of Fences in Pastures

Automatic gates on individual plots can be used to control individual groups of cows divided by yield to grant them access to different areas of pasture with different yields. Control can be implemented by programming the time the animals are in the quarters or by individual remote control by the farmer. The gate system can be combined with an AMS automatic milking system [168,169]. GPS-guided mobile fences are also used, which make a new area of pasture with fresh feed available every pre-programmed time for the grazing cattle herd [41]. Currently, for the most part, the organization of grazing and control of the allocation of plots is limited to the labor-intensive and less efficient conventional fencing system [170].

### 3.2.3. State-of-the-Art Applications and Programmes to Predict Pasture Yields

A new feature is the automatic mowing of the pasture underplanting immediately after the cows have grazed on the plot. There are also more and more computer programmes available to assist the farmer in grazing management, making it possible to predict the start of grazing a month in advance based on the current and predicted weather situation. This allows planning when and for how long the animals will be grazed [41].

### 3.2.4. Automatic Milking System (AMS) at Pasture

On farms where a pasture-based feeding system is used, automatic milking machines are often used for milking. The integration of automatic milking systems (AMS) into pasture-based cattle farming poses new challenges that are very different from those already known in systems where cows are grown in cowsheds. A particular challenge is the grazing of large cattle herds, where more than 50% of the total diet is pasture forage. When an automatic milking system (AMS) is used, animals have to travel considerable distances from the pasture to the milking point [171]. Information reported by Islam et al. [172] shows that cows milked by automatic milking machines had to travel distances exceeding 1.0 km on average, in cases where the farm size was more than 80 ha. Significant distances between the grazing area and the location of the automatic milking machine result in longer intervals between milkings and are associated with increased energy loss by the animals spent on constant movement [173,174]. In addition to the positive sides, frequent

movement can also have negative effects on animal welfare. Travelling long distances increases cortisol levels (an indicator of stress) and can cause gait disturbances or lameness or cause mechanical injuries to the hoof [175].

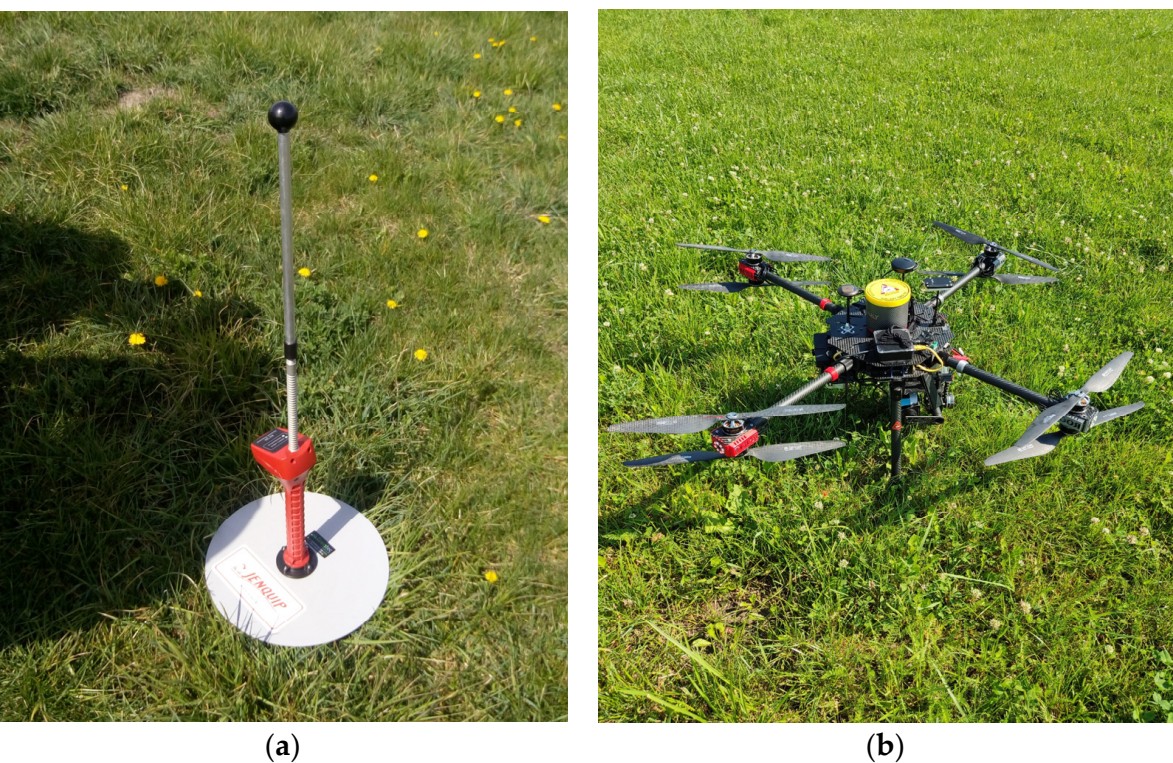

(**a**) (**b**)

**Figure 2.** Innovations applied to grasslands (**a**) Plate meter; (**b**) Drone. (Fot. M. Staniak).

**Table 2.** Innovations in grazing.

| Parameter | Methods of Analysis (Technology) | Use on Grasslands (Application) | References |
|---|---|---|---|
| Yield. Sward filling. | Clipping quadrats (Measurements of the weight of a 1 m² sward). PEAQ sward stick (alfalfa). QMS Sward stick Plate meter (Rising plate meter, Pasture Meters, Quality Plate Meters) Ultrasonic distance Sensors. Spectral sensors in visible or near infrared light. LiDAR and UAS (Unmanned aircraft system). Mutli-View Stereopsis (MVS). | Assessing plant growth rates. Estimating various plant traits such as height, biomass, and ground cover. | [176–184] |
| Sward composition. Sward structure. Identification of rare species in grasslands. Defoliation of sward. | Combination of RGB sensors and hyperspectral sensors. Laser scanning (LiDAR). | Assessing the nutritional value of sward. Decision support for choosing the time of harvesting the sward Identification of valuable natural habitats. Conservation of grazing areas by adjusting the length of grazing time and changing paddocks. | [185–187] |

**Table 2.** *Cont.*

| Parameter | Methods of Analysis (Technology) | Use on Grasslands (Application) | References |
|---|---|---|---|
| Observation and control of animal behavior. | Accelerometer. Pressure sensor. Acoustic sensor. | Detection of estrus and metabolic disorders based on rumination activity. Animal activity and behavior. Selection preferences of plant species in the sward. Frequency of forage intake. Calculation of forage intake per day. Detecting metabolic and digestive disorders from rumination activity. | [188–191] |
| Movement patterns | GPS logger. Pedometer | Individual movement patterns of animals. Movement activity per day. Movement activity of a group of animals or herd. Length of animal grazing and degree of use of pasture area. | |
| Movement on pastures | Automatic opening of gates on paddocks | Control access of grazing animals to paddocks. Preprogrammed times for opening of paddocks. Remote control of gates | [169] |
| Grazing behavior | Virtual fencing | Dynamic control and adjustment of animal grazing. Allocation of new grazing areas by shifting virtual boundaries in GPS. Control of pasture boundaries by active neck collars on the animals. Tracking of animals by GPS positioning system. SMS notification of animal escapes. | [160,192,193] |
| Decision support in sward use. Sward management. | Data-based online tool. | Assessment of sward growth rates. Measurements of pasture productivity. Analyses of fertilization and soil richness. Increasing animal feed intake. Assessment of sward quality in the pasture. Improving milk yields and daily animal increases. | [194] |

*3.3. Innovations to Improve Feed Quality*

3.3.1. Temporary Pastures

The grazing of animals can be carried out not only on permanent grassland but also on temporary pastures. They occupy the soil for one to five years and are made up of graminaceous plants or grasses mixed with legumes and other species. The most common species in this type of pasture include grasses: *Agrostis* spp., *Festuca pratensis*, *Lolium perenne*, and *Dactylis glomerata.* Recently, ryegrass varieties with high growth vigor and high sugar content have been used in temporary grassland swards. The legumes (*Trifolium repens*, *Lotus* spp., and *Medicago sativa)* are rich in protein and can help fix atmospheric nitrogen in soils [195,196].

Green fodder from such pastures, due to the high proportion of valuable grasses and legume species, has a higher protein and sugar content and better digestibility [197]. Temporary pastures are usually used for intensive grazing or grazing with a ration of supplementary roughage. Incorporating temporary pastures into the crop rotation cycle can help increase yields in the short term. It can also change the level and/or quality of soil organic matter and, in the medium term, affect the biological properties of the soil [198].

3.3.2. Multi-Species Pastures MSP

The improvement of degraded pastures is important for increasing pasture herbage yield and animal production. For pasture establishment and renovation, seed mixtures composed of different grass species or grasses with legumes are almost exclusively used, which guarantees the production of large quantities of good quality animal feed [199]. A new aspect in grassland forage production is the addition of herbaceous species naturally occurring in grassland communities to seed mixtures, to obtain multi-species pastures MSP or mixed-herb leys [200,201]. The addition of herbs improves the nutritional value of the pasture sward while maintaining a high and stable yield. *Cichorium intybus* (L.), *Plantago lanceolata* (L.) and *Achillea millefolium* (L.) increase the mineral content, resulting in a better-balanced ratio, improving the animal condition and growth. A well-balanced diet containing herbs in its composition, when used in calves, can influence the subsequent production performance of adult animals [202,203]. Herbs improve the palatability of feed, stimulate digestive processes, and increase the feed intake of animals. Palatability-enhancing species include *Carum carvi* (L.), *Sanguisorba officinalis* (L.), *Daucus carota* (L.), *Pastinaca sativa* (L.), *Rumex acetosa* (L.), and *Salvia officinalis* (L.). The herbs contain specific biologically active substances of tannins, saponins, terpenes, flavonoids, and alkaloids, which can have a positive impact on animal health prevention [204]. Essential oils found in herbs increase palatability and influence the feed intake of animals [18]. Terpenes, flavonoids, and alkaloids have positive effects on cattle gastrointestinal function and health by enhancing the immune system, and antioxidant and antiparasitic effects in the gut [127].

Lambs grazed on pastures containing *C. intybus* showed less infestation with internal parasites and the animals had higher growth rates than animals grazed on pastures without herbs [205]. *Carum carvi* (L.), *Anethum graveolens* (L.), and *Artemisia vulgaris* (L.) have similar effects. These herbs contain tannin compounds and bitters that reduce the incidence of gastrointestinal parasites and also have antidiarrheal effects [206]. A diarrheic effect has been observed when plants such as *Pimpinella anisum* (L.), *Lotus uliginosus* (Schkuhr), and *Anthriscus cerefolium* (L. Hoffm.) are ingested.

The correct percentage of herbs in the feed is important. Through a meta-regression, McCarthy et al. [207] investigated whether there is an optimum inclusion percentage of herb species in a grazing sward to increase milk yield. However, despite a positive relationship between herb percentage in the sward and milk yield, the association between herb percentage and milk yield was non-significant. The authors concluded that continued research investigating management strategies for multispecies swards is needed to determine optimum grazing strategies for multispecies swards in modern pasture-based dairy systems.

An optimally composed multispecies mixture containing herbs in its composition under stress conditions, e.g., drought, can provide a sward yield comparable to a mixture containing only grasses and legumes [208]. Increasing pasture biodiversity through the use of multi-species seed mixtures also has a positive impact on environmental aspects. The deep root system of *Cichorium intybus* (L.) contributes to the utilization of mineral nitrogen from deeper subsoil layers, which is not available in the root system of grasses [209].

## 4. Impact of Grazing on Animal Welfare

The efficient and rational use of pasture is linked to maintaining a balance between pasture productivity and the needs of the animals. The correct preparation of the pasture, in particular ensuring optimal grazing organization, is of key importance for the efficiency of production, and the condition and health of the animals in the herd.

Animal health and welfare are important issues, not only during the grazing season but throughout the year and for all groups of animals. Welfare includes aspects that are relatively easy to measure, such as health, as well as intangible aspects, such as emotions and feelings. In this respect, grazing has both advantages and disadvantages [210,211]. One important aspect of animal welfare is a natural behavior. This includes the need for food, water, and rest, as well as behavioral needs such as movement, social behavior, foraging, and play. Compared to cowshed housing, grazing provides many more opportunities for natural behavior. For cattle, pasture is a natural environment, allowing them to express normal behaviors. It can provide ample comfortable lying space, allowing cows to lie in stretched positions. A literature review by Arnott et al. [212] found that pasture access had benefits for dairy cow behavior, in terms of grazing, improved lying/resting times, and lower levels of aggression. The results of the observations made by Crump et al. [5] show that cattle at pasture had fewer lying bouts but longer lying times, indicating they were more comfortable and less restless. Lying behavior was also more synchronous outdoors, with most of the herd lying at the same time. These results indicate pasture provides a comfortable surface and reduces competition for lying space. Furthermore, cows in pasture walked farther, with potential benefits for their physical health and well-being [213].

Moreover, when given the choice between pasture and indoor housing, cows showed an overall preference for pasture, particularly at night [214]. Overall, grazing has a positive impact on animal health. Regarding health, cows on pasture-based systems had lower levels of lameness, hoof pathologies, hock lesions, mastitis, uterine disease, and mortality compared with cows on continuously housed systems [212,215]. The relatively hard floor in the barn can cause wounds and sores on the knee and heel joints [216,217]. The coefficient of friction required for the free movement of dairy cattle is a minimum of 0.6 [218]. While the coefficient of friction of pasture is higher than 0.8 [219], and the coefficient of friction of floors in conventional barns is generally less than 0.6.

On the other hand, feeding pasture fodder causes large fluctuations in diet composition and makes frequent milking difficult. Both aspects negatively affect welfare, especially if the cows are highly productive. However, animals can be supplemented in the field, but maintaining grazing as the main source of food. There are regions in the world (e.g., subtropical) where it is possible to maintain high quality of cultivated pastures by using temperate or tropical, annual, or perennial species throughout the year. But, of course, there is a need for a good forage planning. In addition, extreme climatic events such as heat waves, extreme rainfall, and prolonged dry spells pose serious challenges to the pasture feeding system [220]. If pasture temperatures exceed 25 °C, this can cause heat stress in animals [221,222], which negatively affects milk yield [223,224]. In addition, outdoors there is an increased risk of infection with specific pathogens such as intestinal worms, lungworms, and liver fluke [225]. Malnutrition and delayed onset of estrous activity postpartum can be observed in cows on pasture. Contact over fences with cows from other farms increases the risk of transmission of infectious diseases such as infectious bovine rhinotracheitis and bovine viral diarrhea. However, in practice, these risks rarely lead to

serious animal health problems. In general, it is easier to prevent grazing defects than to remedy welfare defects of permanently housed animals [210].

Monitoring animal welfare in large pastures can be time-consuming, especially if they are scattered over large areas of semi-natural pasture. Several technologies exist for monitoring animals with different devices that record physiological or behavioral parameters and trigger alarms when the information obtained deviates from the norm [190]. Automatic devices that allow continuous monitoring are ear tags with electronic identification and collars with GPS positioning units (they can assess animal movements and habitat choice and, to some extent, their health and welfare). However, knowledge of the potential impact of digital technology on the monitoring and management of livestock in a grazing system on animal welfare is limited, especially about drones and virtual fences.

## 5. Grazing in Response to Consumer Expectations about Product Quality

EU projections call for a reduction in meat consumption in EU countries by 2050. This would entail a reduction in adverse climatic and environmental impacts. In conjunction with the reduction in meat production and consumption, the consumption of dairy products is forecast to increase. This forecast takes into account the health-enhancing and nutritional qualities of milk and its products as a counter-position to meat, mainly red meat. Milk is a highly nutritious food and a valuable source of minerals, fats, amino acids, and vitamins [226]. However, it is estimated that worldwide only 10–15% of milk production comes from grazing systems [227].

### 5.1. Lipid Fraction

Milk contains about 70% saturated fatty acids and 30% unsaturated acids. Among the latter, a distinction is made between monounsaturated acids (MUFA), which account for about 83% of unsaturated acids, and polyunsaturated acids (PUFA), which account for about 17% [228,229]. Polyunsaturated fatty acids are among the bioactive components of milk. The content of polyunsaturated fatty acids in milk is about 2 g/L, of which linoleic acid (C18:2 n-6) and α-linolenic acid (C18:3 n-3) are the most abundant. Important for human health is the ratio of n-6 to n-3 acids, which should be within 4:1 [230]. Of all the fatty acids found in cow's milk fat, CLA (conjugated linoleic acid) has the greatest health-promoting properties. The importance of this acid is very high, as it exhibits antiatherosclerotic properties, prevents obesity, inhibits the development of certain cancers, and has immune effects.

The fat and fatty acid content of milk is variable and depends mainly on nutrition and less on genetic or physiological factors [231]. Fatty acids in milk can come from two sources: from feed or as products of digestive and metabolic processes in the rumen [229]. Long-chain fatty acids usually originate from the feed and enter the milk via the bloodstream. In contrast, short- and medium-chain fatty acids (C4 to C14 and some C16) are produced in the mammary gland by de novo synthesis from precursors such as acetate and butyrate [232,233]. The levels and types of fatty acids synthesized de novo vary, and this process is controlled by several key genes that are expressed in the mammary gland during milk production [234].

Studies comparing the effect of feeding pasture fodder and TMR on milk fat content have shown significantly higher percentages of health-promoting fatty acids in milk fat from cows fed pasture fodder [235,236]. Moreover, other studies confirm [237,238] that milk from cows fed pasture forage has a lower saturated fatty acid (SFA) content and a higher unsaturated content with a higher proportion of PUFA. Pasture feeding can lower, among other things, the ratio of n-6:n-3 acids in milk fat [239] and increase the content of several beneficial MUFAs and PUFAs (n-3) and their isomers. These include vaccenic acid (trans-11 C18:1), linolenic acid (C18:3), cis-9,trans-11 CLA, trans-11,cis-9-18:2 providing nutrient-rich milk with an improved thrombogenic index. As shown in a study by Kuczyńska [240], pasture feeding can result in an almost three-fold increase in the content of t-accenic, CLA, and α-linolenic acids in milk compared to feeding feed in the form of TMR.

The fatty acid content and structure of cows' milk are influenced by the acids in the roughage. Green forage contains 10 to 12 times more $\alpha$-linolenic acid C18: 3 than cereal grains. Preserved feeds, such as hay, will contain less of it. In addition, the fat contained in pasture greens has a very favorable ratio of n-6 to n-3 acids.

The increase in fatty acids is not only found in milk but also in dairy products such as cheeses [241–246] or butter [241,246,247] produced from milk from cows fed on pasture grass. This can be attributed to the high supply of these nutrients and FA substrates through the fresh pasture, and a high delivery of these FA to the mammary gland which enhances their final concentrations in milk [248,249].

As it turns out, the same beneficial effect on milk quality cannot be obtained by feeding grass brought into the barn. This is because a cow grazing on pasture takes up the forage much more slowly, which has a beneficial effect on rumen function. However, the partial grazing of cows on pasture makes it possible to cover approx. one-third of the animal's daily forage requirements and has a similar effect on milk quality as 8-h grazing.

Pasture feeding influences the taste, texture, and color of the milk, which has a significant impact on the quality and taste of the products made from it. Milk obtained from cows fed on pasture sward has a creamy color (milk from conventionally reared cows is pure white) [250], and all products made from such milk (butter, yoghurt) have a slightly yellow tinge. Similarly, butter and cheeses made from pasture milk were more yellow in color due to higher concentrations of carotene in the milk, had reduced hardness and rancidity ratings at room temperature, and had higher preferences for various attributes such as creaminess, appearance, taste, and color [236,241,246,251,252].

Differences in the composition of the lipid fraction of milk, especially the FA profile in milk from cows fed pasture grass and cows fed TMR, affect milk fat processability (milk fat processability) and the physicochemical and sensory characteristics of the final product.

Butter made from the milk of cows fed on pasture, due to a higher proportion of unsaturated fatty acids, has a softer texture and spreads better [253]. The concentration of unsaturated fatty acids and CLA in milk fat is negatively correlated with the concentration of SFA, and this relationship affects the texture (softness/hardness) of dairy products such as butter [246,247,254,255]. The ratio of palmitic acid (C16:0) to oleic acid (cis-9 C18:1) is referred to as the lubricity index. A higher amount of oleic acid promotes butter to be softer and easier to spread [246].

Butter made from pasture-fed milk is characterized by higher iodine number, acid number and peroxide and thiobarbituric acid (TBA) values. Despite the beneficial effect of pasture feeding on sensory attributes, the increased unsaturated FA content may increase the susceptibility of butterfat to undesirable quality changes such as lipolysis and oxidation [253]. However, the high levels of natural antioxidants, such as tocopherols and carotenoids, transferred to butterfat from forage may also be associated with increased oxidative stability [256].

Moreover, cheeses made from milk from cows fed on pasture grass were characterized by more favorable sensory attributes, such as texture (crispness and granularity), which the authors directly attributed to the variation in FA profiles in the milk [251].

The sensory properties of the milk are influenced not only by the pasture diet itself but also by the botanical composition of the pasture sward. Particular importance is attributed to the herbs and the aromatic oils, volatile fatty acids, and aldehydes they contain, which pass from the blood into the milk and give it its characteristic taste and smell. Comparing the composition of milk from cows grazing on different pastures, it was found that the most beneficial effect on the fatty acid content, both n-6 and n-3, was found to be the feeding of multi-species green forage containing different herbs in their composition.

The fatty acid profile of milk is also particularly favorably influenced by feeding red clover (*Trifolium pratense* L.). Feed rations containing red clover increased the unsaturated fatty acid content of cow's milk, especially $\alpha$-linolenic acid [257]. Red clover owes its properties to its high content of polyphenol oxidase, an enzyme that reduces the process of lipolysis in the rumen, thus increasing the efficiency of transfer of n-3 acids from feed

to milk [258]. For this reason, silage made from a mixture of grass and red clover is recommended for winter feeding, when it is not possible to graze the animals on pasture.

*5.2. Milk Proteins*

Milk is an excellent source of high-quality protein. It contains about 30 different proteins, which occur as casein fractions, whey proteins and fat globule envelopes [226]. These proteins have a high biological value. They are a source of easily digestible essential amino acids. The whey proteins found in milk (the most valuable whey protein is lactoferrin) and the peptides formed from them have anti-inflammatory, bacteriostatic, antioxidant, opioid, anticancer, and antihypertensive properties, among others [259,260]. Specific fragments of milk proteins are thought to be important bioactive peptides with implications for reducing the risk of type two diabetes, obesity development, and high blood pressure [261,262]. Several milk components, including proteins, can interact with immune and neural networks affecting the rate of infection and mood [263]. These bioactive peptides are formed during fermentation by milk starter cultures, found in fermented dairy products and ripened cheeses [264,265]. The ingestion of milk proteins for some individuals might result in the occurrence of an altered or abnormal reaction called cow milk allergy [266].

The feeding regime has been shown to affect milk and protein yields. Both milk protein yield and milk protein content have been found to increase linearly as the proportion of fresh grass in the feeding ration increases [246]. The authors explained it by the linear increase in the propionic acid content in the rumen, which increased milk and protein synthesis, thus the higher protein yield and content. It was also found that milk from pasture-fed cows had a higher protein-to-fat ratio, higher whey protein content (especially β-lactoglobulin and lactoferrin) and better processability [267]. This demonstrates the better bioactive status of pasture-fed milk, especially since lactoferrin and its peptides, are biologically active milk compounds with positive effects on human health.

These differences in milk protein components affect the physicochemical properties of milk, including higher ethanol stability, increased thermal stability, and shorter solidification time of milk from pasture-fed cows [253]. In addition, it was found that cheese made from milk from pasture-fed cows was firmer, and had significantly higher solids content and lower water content compared to cheese made from milk from TMR-fed cows [253]. In addition, significantly higher protein and carbohydrate contents were found in cheese made from milk from pasture [253].

*5.3. Vitamins and Minerals*

Cow's milk is a valuable source of antioxidants. These include vitamin A (retinol), vitamin $D_3$ (cholecalciferol), vitamin E (tocopherol), vitamin $K_2$ (menaquinone), and ß-carotene (provitamin A) [268]. B vitamins are also found in smaller amounts in milk [269]. The pasture is a good source of various vitamins and antioxidants, which are transferred from the forage to the mammary gland and next to the milk. As a consequence, milk from pasture-fed cows has a higher content of ß -carotene, terpenes, lutein, vitamin A (retinol), E (tocopherol), and phytol [270]. In addition, the exposure of animals to the sun while outdoors promotes the synthesis of vitamin D.

Milk is a rich source of many macronutrients, mainly calcium, magnesium, and micronutrients such as selenium, iodine, zinc, copper, and iron. Minerals taken in with feed are not transformed but pass directly into the milk. Therefore, the mineral content of milk depends mainly on the content in feed and the extent to which mineral and vitamin supplements are used in the cows' diet. Milk from cows on pasture had a higher content of calcium and phosphorus [271].

## 6. The Yield Assessment Systems and Grazing Techniques

In recent years, numerous studies have been conducted on the application of remote sensing in estimating and assessing sward growth dynamics in pastures, the availability of forage in particular areas of the pasture, and even the vegetation structure in the sward [272].

Active optical sensors calculate the normalized differential vegetation index (NDVI) in real-time by measuring the reflectance in near-infrared and red light and in this way, the yield and quality of the pasture sward are calculated [273]. Measurements can be taken manually with an NDVI sensor by taking random measurements on selected areas, or such devices can be mounted on tractors or vehicles, allowing more measurements to be taken over a larger area of pasture.

Unmanned aerial vehicles (drones or UAVs for short) are already widely used for monitoring field crops and permanent grasslands [274]. These devices are equipped with cameras that create spectral images of grasslands and are transmitted in real-time by radio waves creating images on the computer monitors of farmers monitoring the field crop and grassland area. Current research focuses on the validation of drone-derived images for estimating sward weight and quality in different grassland habitats through the use of algorithms and predictive modelling of yield and quality [182,275]. These results allow the generation of maps that inform the spatial variability and quality of the pasture sward. In addition, remote sensing-based monitoring using Sentinel and Lidar satellite systems provides some information on sward yield weight, forage quality, dynamics and rates of daily sward growth, and even potential estimates of pasture grazing animal performance in terms of milk production and meat gains [276]. Such monitoring of pastures and estimation of their yield potential even allows for a quick response and proactive adjustment of the number of grazing animals to the pasture area or the choice of when to swath the grassland [187]. Drones with on board cameras and image analyzers have great potential for use in pastoralism, monitoring the herd, and searching for lost animals [277].

The latest research topics are technologies for monitoring and controlling the movement of animals and their allocation to particular quarters according to their performance. Grazing management requires the farmer to precisely control where the animals are, as well as their behavior in the pasture, i.e., how long they graze and what physical activity they have performed. Technical innovations for animal monitoring are often limited to herd management in free-stall barns in the form of automated milking systems, pre- and post-milking animal weighing systems and electronic cow identification and monitoring systems when more feed is rewarded for cows with higher milk yields [278]. In free-stall barns, monitoring of animal location, behavior, feed intake frequency, frequency of approach to milking equipment, start and end of heat period, and current changes in health status are carried out using collars and transponders. The monitoring devices include accelerometers, thermometers, pressure gauges, microphones, and rumen activity sensors. These are attached to the animals' bodies via ear tags, leg recorders, or placed in collars [188,190,191]. Any deviation from the normal behavior of individual animals can be quickly noticed by the keeper, who is informed via SMS messages on the phone. Fitting the animals with a GPS satellite positioning receiver collar enables detailed information to be obtained on the rate and manner of their movement and behavior in the herd [279,280].

Virtual fencing (VF), already described in more detail in the previous topic (see Section 3.2.1), is an innovation in digital systems for controlling animal feed allocation in pastures without physically setting up posts, wires, and other permanent fences [160]. Virtual fencing allows grazing animals to be directed to new plots at a specific time and kept away from plots that have just been used and which need to be given adequate time to regrow. The system does not adversely affect animal behavior, welfare, or performance (no negative impact on live weight gain or milk yield) [192,281]. Virtual fencing can also be an opportunity to introduce grazing animals (sheep) in previously unused areas, protected habitats where physical fencing is prohibited or with difficult access such as riparian meadow areas [192] and moorland in Scotland [160].

Despite the availability of many support tools, still, a small percentage of farmers choose to use them on their farms [282]. This is likely because innovative decision support systems and devices have not yet convinced farmers of the economic benefits and compliance with animal welfare practices. The cost of their use and maintenance on the farm can also have a large impact on the level of use of innovations [283].

Monitoring and collecting agronomic grassland information data using remote sensing generates large amounts of data that need to be presented in easy-to-understand and readable graphs, maps, and animations developed in simple software and apps available on phones. Such tools in the present and future will provide farmers with comprehensive knowledge and assistance for grazing management [284]. Farmers' decision support systems have great potential to facilitate record-keeping and tracking of the agricultural production process for regulators and consumers of agricultural products [285].

All these available tools can help to better understand grazing, manage stocking density and animal load on the pasture and, most importantly, better understand the impact of the grazing process on the environment and climate change.

## 7. Conclusions

In most regions of the world, grazing plays an important role in milk production. Despite the importance of grazing, current trends in animal husbandry in Europe are causing a decline in the popularity of feeding cattle, especially dairy cows, on pasture. Grazing provides a range of ecosystem services, including regulating, storing, and purifying biogenic water, regulating plant nutrient cycles, increasing C sequestration, while also reducing greenhouse gas emissions. Livestock grazing is the most important factor shaping and stabilizing the biodiversity and botanical composition of plants in pastures. Important advantages of pasture-based feeding of dairy cows include the high-quality and health-promoting properties of milk and milk products, which translate into health benefits for consumers. The beneficial effects of pasture feeding on animal health, condition and welfare are also not insignificant. Available organizational innovations help to manage livestock grazing more effectively. By automating certain tasks and introducing new technologies, grazing efficiency, the quality of the products obtained and working conditions are improved.

**Author Contributions:** Conceptualization, B.W., W.Z. and M.S.; writing—original draft preparation, B.W., W.Z. and M.S.; writing—review and editing, B.W., W.Z. and M.S.; visualization, B.W., W.Z. and M.S. All authors have read and agreed to the published version of the manuscript.

**Funding:** This research received no external funding.

**Institutional Review Board Statement:** Not applicable.

**Data Availability Statement:** Not applicable.

**Conflicts of Interest:** The authors declare no conflict of interest.

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
