# Peer review of "Challenges of Pasture Feeding Systems—Opportunities and Constraints"

_agriculture, doi:10.3390/agriculture13050974_

Round 1

Reviewer 1 Report

Please, see attached file.

Author Response

Dear Reviewer,

Thank you for your effort involved in preparing the review of our manuscript “Challenges of Pasture Feeding Systems - Opportunities and Constraints”. The feedback has been invaluable in improving the content and presentation of the paper.

We have revised our manuscript according to all of reviewers' comments, as explained after the word ‘Response:’ in the point-by-point responses below.

RESPONSE TO REVIEW:

Point 1: L29. It seems too low the value 9%, please verify and add a reference.

Response: Thank you for your comment. This value comes from the publication by De Haan, C.; Steinfeld, H.; Blackburn, H. Livestock and the environment: Finding a balance. Rome: European Commission Directorate-General for Development, Development Policy Sustainable Development and Natural Resources, 1997; p. 115. (item 1 in the reference list) and is cited at the end of this paragraph.

Point 2: L40. In temperate regions, most of the year animals are indoor, mainly in regions with snow during the winter. Grass grows for most of the year in tropical and in subtropical regions, with a higher proportion of dairy production systems being pasture-based in these regions!

Response: Thank you for your comment. The sentence was changed on: Pasture-based dairy production systems are mainly found in temperate regions, where green forage is the cheapest feed used in milk production.

Point 3: L52. What do you mean with “forage weight before grazing”? The term “biomass” seems better. Sward height “pre-“ and pos- grazing seems important.

Response: Thank you for your comment. The sentence was changed on: For example, rotation length, biomass weight before grazing [11] and sward height pre- and pos- grazing [12] can affect grass quality, as well as grass supply.

Point 4: L126. Please, provide a reference for this statement. The same is true for “rotatinuous stocking”.

Response: Thank you for your comment. The words “rotatinuous stocking” were added and two references were cited: Apfelbum, S.L.; Thompson, R.; Wang, F.; Mosier, S.; Teague, R.; Byck, P. Vegetation, water infiltration, and soil carbon response to Adaptive Multi-Paddock and Conventional grazing and Southeastern USA ranches. J. Environ. Manag. 2022, 308, 114576. 1185 https://doi.org/10.1016/j.jenvman.2022.114576

Savian JV, Schons RMT, de Souza Filho W, Zubieta AS, Kindlein L, Bindelle J, Bayer C, Bremm C, Carvalho PCF. 'Rotatinuous' stocking as a climate-smart grazing management strategy for sheep production. Sci Total Environ. 2021 Jan 20;753:141790. doi: 10.1016/j.scitotenv.2020.141790. Epub 2020 Aug 22. PMID: 32890869.

Point 5: L162. It would be better to indicate the % of reduction in the emission per kg of product, since 56% of increase is a high value.

Response: Thank you for your comment. The reduction in the emission in % per kg of the product was indicated.

Point 6: L180. This statement is not clear for me: “Grazing has several effects on the environment but the most evident is nutrient loss.” What the authors mean by “nutrient loss” via grazing? Several studies have reported that agroecosystems productivity can be enhanced via nutrient cycling by grazing animals (e.g. Farias et al., 2020, Agronomy for Sustainable Development, 40:39; Lemaire et al., 2023, Agronomy, 13, 982; Thompson et al., 2023; Animal Frontiers, Volume 13, Issue 2, April 2023, Pages 28–34)! The multiple benefits and positive impact of grazing, from the ability of domestic herbivores to recycle and transfer mineral nutrients across agroecosystems, is much more important and generalizable than some nutrient loss via manure concentration. Further, the inclusion of grazing animals in agroecosystems can reduce the necessity for external inputs (see the references cited above as examples).

Response: Thank you for your comment. It was added “reduction of nutrient loss”

Point 7: L299. “Characterized”

Response: Thank you for your comment. It was improved.

Point 8: Topic “grazing systems”: According to Allen et al., 2010 (Grass and Forage Science, 66, 2–28) the term ‘stocking’ is preferred to ‘grazing’ (i.e. ‘stocking method’ vs. ‘grazing method’) because grazing refers to the consumption of standing forage, whereas it is the method of stocking grazing animals that allows manipulation of how, when, what and how much the animals graze. It seems in this topic that methods (e.g., rotational, continuous) are confused with systems (e.g., silvopastoral).

Response: Thank you for your comment. grazing systems” were replaced with “stocking method”.

Point 9: L364. More important than the stocking method, i.e., continuous x rotational, is the grazing intensity! This must be clear.

Response: Thank you for your comment. This information was added to the text.

Point 10: L482. Again this statement given much emphasis in the cost for grazing animals. Grazing is the cheaper way to feed domestic herbivores, and structures to maintain animals indoors have a higher cost than to maintain in pastures even with different paddocks.

Response: Thank you for your comment. This information was supplemented in the text.

Point 11: L635. However, animals can be supplemented in the field, but maintaining grazing as the main source of food. There are regions in the world (e.g., subtropical) where is possible to maintain high quality of cultivated pastures by using temperate or tropical, annual, or perennial species throughout the year. But, of course, there is a need for a good forage planning.

Response: Thank you for your comment. This information was supplemented in the text.

Point 12: L845. Previous “topic” seems more adequate than chapter.

Response: Thank you for your comment. The word “chapter” was replaced to “topic”.

All authors have read and approved the revised manuscript. We hope that our resubmission is now suitable for inclusion in Agriculture and we look forward to hearing from you.

Yours sincerely,

Reviewer 2 Report

Report on paper no: agriculture-2358533

Challenges of Pasture Feeding Systems - Opportunities and Constraints

By Barbara Wróbel, Waldemar Zielewicz and Mariola Staniak

 General Comment

This paper (review) addresses the important study area concerning the grazing system, with more reference to dairy cows. The paper provides several and updated information on the most important aspects in this area, citing numerous works that address these aspects. It is well written, clear and addresses the topic reporting in a sufficiently clear and complete manner on the aspects it proposed to deal with. In general, the paper is relevant for the field and presented in a well-structured manner.

I recommend accepting the paper after a minor revision

General remarks:

I did not find any general remarks to make. Just two personal opinions: the authors probably could have made an effort to elaborate more critically on some of the information and results of other authors in the review. The second: perhaps there is an excess of references in the Polish language, which is not easily translatable, I imagine

Detailed remarks:

Lines 136-139: the authors mention the concept of degradation of grassland. At least when they mention certain practices that determine it, they could give some examples..

Paragraph 2.3: the title does not seem perfectly in line with what was then written

Line 202: please add a reference about what is stated

Lines 493 and 500: is it possible to find a reference in English?

Line 594: the authors speak of an “excessive amounts”, but they could be a little more precise, also taking into account that the cited work is not in English

Line 783: please add a reference about what is stated

Author Response

Dear Reviewer,

Thank you for your effort involved in preparing the review of our manuscript “Challenges of Pasture Feeding Systems - Opportunities and Constraints”. The feedback has been invaluable in improving the content and presentation of the paper.

We have revised our manuscript according to all of reviewers' comments, as explained after the word ‘Response:’ in the point-by-point responses below.

RESPONSE TO REVIEW:

Point 1. Lines 136-139: the authors mention the concept of degradation of grassland. At least when they mention certain practices that determine it, they could give some examples.

Response: Thank you for your comment. It was clarified.

Point 2: Paragraph 2.3: the title does not seem perfectly in line with what was then written

Response: Thank you for your comment. The title was changed.

Point 3: Line 202: please add a reference about what is stated

Response: Thank you for your comment. The reference was added.

Point 4: Lines 493 and 500: is it possible to find a reference in English?

Response: Thank you for your comment. The following reference in English was found:

Mancuso, D.; Castagnolo, G.; Porto, S.M.C. Cow Behavioural Activities in Extensive Farms: Challenges of Adopting Automatic Monitoring Systems. Sensors 2023, 23, 3828. https://doi.org/10.3390/ s23083828

Point 5: Line 594: the authors speak of an “excessive amounts”, but they could be a little more precise, also taking into account that the cited work is not in English

Response: Thank you for your comment. It was difficult to find the information about optimal amount of herbs. We have added information on the results of a study aimed at finding the optimum inclusion percentage of herb species in a grazing sward.

Point 6: Line 783: please add a reference about what is stated

Response: Thank you for your comment. The reference was added.

All authors have read and approved the revised manuscript. We hope that our resubmission is now suitable for inclusion in Agriculture and we look forward to hearing from you.

Yours sincerely,

Reviewer 3 Report

 The manuscript provides a comprehensive review of the opportunities presented by pasture grazing in the context of climate change. The article presents convincing arguments to support the idea that pasture grazing is not only beneficial for producing high-quality products and promoting animal welfare, but it could also result in fewer carbon emissions than indoor farming. The paper presents compelling evidence that grazing livestock on pasture can reduce greenhouse gas emissions and improve soil health, which are critical considerations in the fight against climate change. This review would be an invaluable resource for readers interested in pasture-based livestock production or those advocating for its advancement. The writing style is engaging, well-structured, and easy to follow, which makes it an excellent reference for those seeking to deepen their understanding of this important topic. Some minor suggestions are as follows.  

Line 33-36: Information on declining trends of pasture should be moved to the appropriate paragraph discussing the challenges of pasture grazing, which is in lines 84-90. The current paragraph can start with discussing the status of pasture grazing and its importance.

Line 39: Add the reference for this statement.

Line 40-43: To improve coherence, these lines can be placed before the second paragraph (lines 33-39).

Line 105-106: The major source of methane emission is from ruminal fermentation through belching, not excretion. Rephrase the sentence to focus on this instead.

Line 373: Rational grazing?

Line 459: SPS?

Line 632: Add the unit of frictional force for clarity.

Line 717: Add a reference to support the statement about pure white milk.

Line 801: Rephrase this sentences as a heading instead of a complete sentence.

It would be helpful to have a concluding paragraph or summary of the review.

Author Response

Dear Reviewer,

Thank you for your effort involved in preparing the review of our manuscript “Challenges of Pasture Feeding Systems - Opportunities and Constraints”. The feedback has been invaluable in improving the content and presentation of the paper.

We have revised our manuscript according to all of reviewers' comments, as explained after the word ‘Response:’ in the point-by-point responses below.

RESPONSE TO REVIEW:

Point 1: Line 33-36: Information on declining trends of pasture should be moved to the appropriate paragraph discussing the challenges of pasture grazing, which is in lines 84-90. The current paragraph can start with discussing the status of pasture grazing and its importance.

Response: Thank you for your comment. The information has been moved.

Point 2: Line 39: Add the reference for this statement.

Response: Thank you for your comment. The reference was added.

Point 3: Line 40-43: To improve coherence, these lines can be placed before the second paragraph (lines 33-39).

Response: Thank you for your comment. The sentences have been moved.

Point 4: Line 105-106: The major source of methane emission is from ruminal fermentation ethrough belching, not excretion. Rephrase the sentence to focus on this instead.

Response: Thank you for your comment. The sentence was rephrased

Point 5: Line 373: Rational grazing?

Response: Thank you for your comment. It should be “rotational” grazing

Point 6: Line 459: SPS?

Response: Thank you for your comment. SPS is an abbreviation for the silvopastoral systems.

Point 7: Line 632: Add the unit of frictional force for clarity.

Response: Thank you for your comment. It was changed to “coefficient of friction” which has no unit.

Point 8: Line 717: Add a reference to support the statement about pure white milk.

Response: Thank you for your comment. The reference was added.

Point 9: Line 801: Rephrase this sentences as a heading instead of a complete sentence.

Response: Thank you for your comment. The sentence was rephrased.

Point 10: It would be helpful to have a concluding paragraph or summary of the review.

Response: Thank you for your comment. A summary has been completed.

All authors have read and approved the revised manuscript. We hope that our resubmission is now suitable for inclusion in Agriculture and we look forward to hearing from you.

Yours sincerely,
